# Clinical outcomes in patients with atrial fibrillation treated with DOACs in a specialized anticoagulation center: Critical appraisal of real-world data

**Carla Moret**[1,2]*, **René Acosta-Isaac**[1,2], **Sergi Mojal**[2], **Mariana Corrochano**[1,2], **Blanca Jiménez**[1,2], **Melania Plaza**[2], **Juan Carlos Souto**[1,2]

**1** Thrombosis and Hemostasis Unit, Hospital de la Santa Creu i Sant Pau, Barcelona, Spain, **2** Institut de Recerca de l'Hospital de la Santa Creu i Sant Pau, Institut d'Investigacions Biomèdiques IIB-Sant Pau, Barcelona, Spain

* carlamoretp@gmail.com

**Data Availability Statement:** All relevant data are within the manuscript and its Supporting Information files.

## Abstract

### Aims

Direct oral anticoagulants (DOAC) are progressively replacing vitamin K antagonists in the prevention of thromboembolism in patients with atrial fibrillation. However, their real-world clinical outcomes appear to be contradictory, with some studies reporting fewer and others reporting higher complications than the pivotal randomized controlled trials. We present the results of a clinical model for the management of DOACs in real clinical practice and provide a review of the literature.

### Methods

The MACACOD project is an ongoing, observational, prospective, single-center study with unselected patients that focuses on rigorous DOAC selection, an educational visit, laboratory measurements, and strict follow-up.

### Results

A total of 1,259 patients were included. The composite incidence of major complications was 4.93% py in the whole cohort vs 4.49% py in the edoxaban cohort. The rate of all-cause mortality was 6.11% py for all DOACs vs 5.12% py for edoxaban. There weren't differences across sex or between Edoxaban reduced or standard doses. However, there were differences across ages, with a higher incidence of major bleeding complications in patients >85 years (5.13% py *vs* 1.69% py in <75 years).

### Conclusions

We observed an incidence of serious complications of 4.93% py, in which severe bleeding predominated (3.65% py). Considering our results, more specialized attention seems necessary to reduce the incidence of severe complications and also a more critical view of the

**Funding:** The Hemostasis and Thrombosis Unit and the IIB-Sant Pau have received funding from Daiichi-Sankyo to develop and maintain the MACACOD registry (Clinical Application Model of Direct Oral AntiCoagulants). The funders had no role in study design, data collection and analysis, decision to publish, or preparation of the manuscript.

**Competing interests:** I have read the journal's policy and the authors of this manuscript have the following competing interests: JCS has received financial support for travel, accommodation, or expenses from Laboratorios Rovi, Leo Pharma, Baxter, Sanofi, Boehringer Ingelheim, Pfizer, Bristol Myers Squibb, Roche, Daiichi-Sankyo, Stago Laboratories and performs an advisory role for Devicare. This does not alter our adherence to PLOS ONE policies on sharing data and materials. CM, RA, SM, MP, MC, BJ have declared that no competing interests exist.

literature. Considering our results, and our indirect comparison with many real-world studies, more specialized attention seems necessary to reduce the incidence of severe complications in AF patients receiving DOACs.

## Introduction

In the last decade, direct oral anticoagulants (DOACs) have been progressively replacing vitamin K antagonists in the prevention of thromboembolism in patients with atrial fibrillation (AF) due to their superior efficacy, safety, and ease of use. However, their real-world clinical outcomes appear to be rather contradictory in terms of major complications. Some studies have reported even fewer complications than in the pivotal randomized controlled trials (RCTs) [1–4]. Most of these excellent results come from retrospective cohorts with data stored in giant databases, mainly from insurance services [5–10] or large prospective registries, some international [11, 12] and some from individual countries [13, 14], that, in some cases, may underestimate serious complications. However, there are also reports from large retrospective nationwide [15, 16] or multicenter cohorts [17] describing an incidence of major complication, especially for hemorrhagic complications.

None of the published studies that compare the incidence of serious complications offer information of DOAC management. These aspects could help to explain the differences mentioned above. Other variables that could explain such differences are the different populations' characteristics, such as age or comorbidities: for example, some are limited to inpatients [9, 18, 19]. Many of the large-scale studies, with tens or hundreds of thousands of patients [5, 7, 8, 10–12, 14, 20–22] report an incidence of mortality much lower than expected in patients with AF [23–25]. In contrast, most studies with a high incidence of serious complications also report significantly higher mortality rates, in line with those expected for these populations [16, 17, 26–30].

We present the results of a clinical model for the management of DOACs in real clinical practice that could be useful to understand these discrepancies.

The MACACOD project started in 2019 as a model to ensure the application of the international DOAC guidelines in routine clinical practice [31]. The main objective was to reduce major complications by following strict patient education protocols, laboratory monitoring, selection of DOAC dose, assessment of drug interactions, and an exhaustive registry of complications during follow-up. The patients were unselected, included at the moment of DOAC prescription, and most of them received edoxaban.

We also carried out an indirect comparison with other published registries.

## Materials and methods

The MACACOD project (Clinical Application Model for Direct Oral Anticoagulants, NCT04042155) is an ongoing, observational, prospective, single-center study with unselected patients treated at the Hemostasis and Thrombosis Unit of a University Hospital.

Patients over 18 years of age with a diagnosis of AF, receiving anticoagulant therapy with DOACs for the prevention of stroke or systemic embolism due to high thromboembolic risk ($CHA_2DS_2$-VASc score [32] more than one) were included. The study was conducted according to the Declaration of Helsinki. Before starting the study, the Ethics Committee of the Hospital de la Santa Creu i Sant Pau approved the study protocol (IIBSP-ACO-2018-21) and all the included patients gave written, signed and dated informed consent.

Patients were treated with DOACs following the indications in the 2016 Therapeutic Positioning Report UT_ACOD / V5 / 21112016 from the Spanish Agency for Medicines and Health Products [33]. Patients were followed up until DOAC treatment was discontinued, the patient died or decided to end his/her participation in the study.

## Inclusion criteria

- Known hypersensitivity or specific contraindication to the use of VKA.

- History of intracranial hemorrhage (ICH), before starting oral anticoagulant therapy (OAT) or during OAT with VKA.

- Ischemic stroke with clinical and/or neuroimaging high-risk criteria for ICH (HAS-BLED score [34] of three or more, grade III-IV leukoaraiosis, multiple cortical microbleeds).

- History of serious arterial thromboembolic event while on treatment with VKA despite good INR control (time in therapeutic range, TTR >65%).

- Poor INR control while on treatment with VKA despite good therapeutic compliance (TTR <65%).

- Unable to access conventional INR monitoring.

## Exclusion criteria

- Current treatment with VKA and good INR control.

- Atrial fibrillation with severe heart valve disease.

- Patients unable to guarantee collaboration: significant cognitive impairment, alcoholism or psychiatric disorders, unsupervised.

- Situations in which OAT is contraindicated: pregnancy, severe acute bleeding in the previous month, recent (ten days prior) or planned CNS surgery, severe liver or kidney disease with creatinine clearance according to Cockcroft-Gault formula (CrCl) less than 15 mL/min, severe or uncontrolled hypertension, hereditary or acquired altered hemostasis (coagulation, fibrinolysis, platelet function) with a significant risk of bleeding.

## Inclusion in the study and data collection

Recruitment of patients began in July 2019 and data was collected until March 2022. The scheduling of visits, follow-up, and blood tests were the same as for patients not included in the study (usual clinical practice). They were organized chronologically as follows:

- First medical and specialized nurse visit: baseline clinical information values were collected for each patient. Information on risk of thromboembolic events (CHA$_2$DS$_2$-VASc), risk of bleeding (HAS-BLED), previous history of thrombosis or bleeding, possible contraindications, laboratory values including kidney function, CrCl and liver function, and selection and dosage of DOAC were also recorded. The selection of the drug to be prescribed was independent of inclusion in the study. Dosage was based on the technical datasheet for each drug and international guidelines [31, 35].

- Educational session: In the first month of DOAC therapy, patients received an individual or group educational session on anticoagulation to learn general concepts, specific information

on DOACs, risks when taking DOACs, drug interactions, the importance of adherence, response to complications, management in case of preoperative or invasive procedures, management in case of missing a dose or double doses, and help contact phone numbers.

- Follow-up visits: A face-to-face visit took place approximately 1 month after starting DOACs for all patients. There were quarterly visits for patients with progressive renal failure, biannual for those at high thrombotic risk, annual for those with medium or low risk, and on-demand visits in case of complications, surgical or invasive procedures, or new potential pharmacological interactions. At each visit, we collected data about tolerance and adherence to the drug, weight, new diagnoses and medication, laboratory data, and thrombotic or bleeding complications since the last visit.

**Blood tests.**   In all patients who received edoxaban, apixaban, or rivaroxaban, anti-FXa (factor Xa) activity was monitored (HemosIL, Liquid Anti-FXa, Werfen) at trough (pre-dose) and peak (2 hours' post-dose) levels, before the first follow-up visit. In 387 of the patients on edoxaban, the plasma concentration of the drug was also measured before and after the dose (HemosIL, Liquid Anti-FXa calibrated for edoxaban) to demonstrate that anti-FXa levels highly correlated with plasma concentrations of edoxaban [36].

**Outcome measures.**   Treatment outcomes were divided into effectiveness and safety. The primary effectiveness outcome comprised ischemic stroke and systemic embolism as major thrombotic complications and all-cause death during DOAC treatment, similar to the pivotal trials. The outcome *clinically relevant non-major thrombosis* (CRNMT) included myocardial infarction, transient ischemic attack, venous thromboembolism and superficial thrombophlebitis. Safety outcomes related to bleeding events were defined according to the Bleeding Academic Research Consortium (BARC) Scale [37]: *major hemorrhagic complications* imply BARC type three to five and *clinically relevant non-major bleeding* (CRNMB) implies BARC type two. A composite outcome of major thromboembolic and major hemorrhagic complications was added to evaluate the overall risk of each DOAC.

## Statistical analyses

Descriptive analysis was conducted. A bivariate analysis was performed to assess differences in baseline characteristics between DOAC group, type of patient, age and sex. Chi-square test or Fisher exact test were used as appropriate to compare categorical variables, and non-parametric Mann-Whitney U test was used to compare continuous variables. Incidence rates and their 95% confidence intervals (exact method) were calculated for the different types of complications and mortality, and were compared between groups with Chi-square test. Finally, a Cox proportional hazards survival analysis was used to assess the relationship between baseline factors and the first occurrence of each type of complication. In all analyses, p-values less than 0.05 were considered statistically significant. Analysis was performed with SPSS 26.0 (IBM Corp.) and R 4.1.1 (R Core Team).

## Results

A total of 1,259 patients were included in the MACACOD registry. After excluding the patients who were not using DOACs for stroke prevention in AF or had not had follow-up, 817 patients (75.3% edoxaban, 16.0% apixaban, 7.1% dabigatran, and 1.6% rivaroxaban) were eligible for analysis (**Fig 1**).

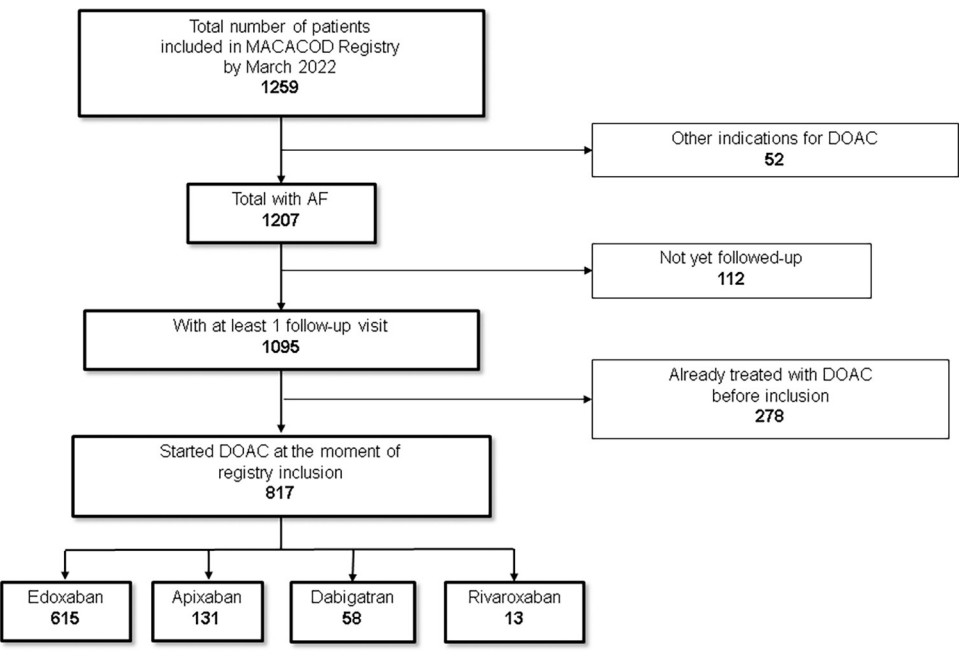

**Fig 1. MACACOD patients' flowchart.**

## Baseline characteristics

Clinical characteristics of patients, by DOAC, are provided in **Table 1**, with statistical comparisons between edoxaban vs apixaban and edoxaban vs dabigatran. We avoided comparisons with the rivaroxaban group due to the low number of patients.

Mean patient age was 77.7 years (SD: 8.3) and was lower in the dabigatran cohort (74.6 years, SD 10.1, p<0.05) than the edoxaban or apixaban cohort (78.0 and 77.7 years, respectively). We did not find significant sex differences among DOAC groups (52.9% male overall). There was a higher proportion of anticoagulant-naive patients in the apixaban and dabigatran cohorts (39.7% and 63.8%, respectively) than in the edoxaban cohort (14.6%). The mean $CHA_2DS_2$-VASc and HAS-BLED scores were higher among those using apixaban (4.37 points, SD 1.78, and 2.45 points, SD 0.84) as reflected by the CCI (1.93 points, SD 1.65). The laboratory characteristics were similar among the different DOAC cohorts. We found significant differences in baseline hemoglobin levels, being lowest in apixaban (126.6g/L) and highest in the dabigatran cohort (139.6g/L). The same occurred for renal function 60.1 mL/min in apixaban and 62.9 mL/min in edoxaban. The CrCl in the dabigatran cohort was higher than in the edoxaban cohort and the dabigatran cohort had the highest history of previous stroke (24.1%).

We also found significant differences between edoxaban and apixaban in anti-FXa activity trough level (0.10 and 0.62 UI/Ml, P<0.01) and the trough-peak difference (1.14 and 0.67 UI/ml, P<0.01).

Between patients taking both edoxaban doses, those on the 30mg were older (81.4 vs 75.9 years), with higher $CHA_2DS_2$-VASc score (4.42 vs 3.69) and with worse mean CrCl (44.3 vs 74.3 mg/dL) but without statistically significant differences except for anti-FXa activity trough, peak and its difference (view Table 1).

## Main outcome analyses

The median follow-up time was 13.7 months. In total there were 13 major thrombotic events, 37 major hemorrhagic events, and 62 deceased patients. The corresponding event rates were

**Table 1. Baseline clinical characteristics according to DOAC type.**

| | Overall | Edoxaban* | Edoxaban 30 mg | Edoxaban 60 mg | Apixaban* | Dabigatran* | Rivaroxaban* |
|---|---|---|---|---|---|---|---|
| **N** | 817 | 615 | 233 | 382 | 131 | 58 | 13 |
| **Age (years)**; mean (SD) | 77.7 (8.3) | 78 (7.8) | 81.4 (7.0) | 75.9 (7.6) | 77.7 (9.3) | 74.6 (10.1) † | 76.2 (7.8) |
| **Sex (male)**; n (%) | 432 (52.9%) | 316 (51.4%) | 86 (36.9%) | 230 (60.2%) | 74 (56.5%) | 37 (63.8%) | 5 (38.5%) |
| **Naive patients**; n (%) | 185 (22.6%) | 90 (14.6%) | 31 (13.3%) | 59 (15.4%) | 52 (39.7%) ‡ | 37 (63.8%) ‡ | 6 (46.2%) ‡ |
| **CHA$_2$DS$_2$-VASc**; mean (SD) | 4.02 (1.57) | 3.97 (1.52) | 4.42 (1.49) | 3.69 (1.46) | 4.37 (1.78) † | 3.78 (1.63) | 4.15 (1.46) |
| **HAS-BLED**; mean (SD) | 2.17 (0.80) | 2.10 (0.76) | 2.14 (0.81) | 2.08 (0.73) | 2.45 (0.84) ‡ | 2.28 (1.04) † | 2.23 (0.60) |
| **CCI**\*\*; mean (SD) | 1.54 (1.47) | 1.46 (1.44) | 1.67 (1.46) | 1.33 (1.41) | 1.93 (1.65) ‡ | 1.43 (1.33) | 1.69 (1.25) |
| **BMI (kg/m$^2$)**; mean (SD) | 27.20 (5.17) | 27.02 (4.87) | 24.54 (3.99) | 28.53 (4.75) | 27.29 (5.64) | 28.01 (5.32) | 30.86 (10.02) |
| **Hemoglobin (g/L)**; mean (SD) | 132.6 (17.8) | 133.2 (16.6) | 127.5 (16.0) | 136.6 (16.0) | 126.6 (21.3) ‡ | 139.6 (18.3) ‡ | 133.2 (15.7) |
| **Bilirubin (mg/dL)**; mean (SD) | 12.9 (6.2) | 12.8 (5.9) | 12.5 (5.7) | 13.0 (6.1) | 13.2 (7.2) | 14.6 (6.9) | 9.8 (3.3) |
| **AST (mg/dL)**; mean (SD) | 21.0 (8.5) | 21.1 (8.1) | 21.3 (8.6) | 20.9 (7.8) | 21.2 (10.5) | 19.2 (6.5) | 23.9 (9.9) |
| **ALT (mg/dL)**; mean (SD) | 19.1 (11.7) | 18.8 (11.0) | 17.5 (1.04) | 19.6 (11.3) | 20.3 (16.1) | 19.4 (8.9) | 22.5 (8.7) |
| **CrCl (mL/min)** \*\*\*; mean (SD) | 63.8 (25.5) | 62.9 (22.5) | 44.3 (10.9) | 74.3 (20.1) | 60.1 (31.1) ‡ | 77.4 (30.1) ‡ | 81.2 (46.4) |
| **CrCl**\*\*\* < **30 mL/min**; n (%) | 34 (4.20%) | 13 (2.10%) | 13 (5.6%) | 0 (0.0%) | 19 (14.5%) ‡ | 1 (1.7%) ‡ | 1 (7.7%) |
| **Anti-FXa act. trough** (UI/mL); med [25–75] | 0.11 [0.06–0.20] | 0.10 [0.06–0.15] | 0.09 [0.05–0.13] | 0.11 [0.06–0.16] φ | 0.62 [0.40–0.94] ‡ | - | 0.21 [0.07–0.60] |
| **Anti-FXa act. peak** (UI/mL); med [25–75] | 1.26 [0.92–1.68] | 1.25 [0.91–1.64] | 0.99 [0.78–1.17] | 1.51 [1.17–1.85] φ | 1.26 [0.95–1.87] | - | 2.23 [1.71–2.76] |
| **Anti-FXa act. difference**; med [25–75] | 1.07 [0.76–1.48] | 1.14 [0.81–1.54] | 0.89 [0.69–1.07] | 1.40 [1.07–1.72] φ | 0.67 [0.46–0.96] ‡ | - | 1.96 [1.64–2.33] |
| **Antiplatelets**; n (%) | 75 (9.2%) | 42 (6.8%) | 13 (5.6%) | 29 (7.6%) | 21 (16%) ‡ | 11 (19%) ‡ | 1 (7.7%) |
| **History of stroke**; n (%) | 110 (13.5%) | 72 (11.7%) | 32 (13.7%) | 40 (10.5%) | 23 (17.6%) | 14 (24.1%) † | 1 (7.7%) |
| **History of major bleeding**; n (%) | 114 (14.0%) | 73 (11.9%) | 30 (12.9%) | 43 (11.3%) | 28 (21.4%) ‡ | 12 (20.7%) | 1 (7.7%) |

* Standard and reduced doses

** CCI: Charlson Comorbidity Index

*** CrCl: creatinine clearance from Cockcroft-Gault formula (mL/min).

† Statistically significant vs edoxaban group with P<0.05.

‡ Statistically significant vs edoxaban group with P<0.01.

φ Statistically significant vs edoxaban 30mg group with P<0.01.

1.28% patient-years (py) (95% confidence interval [CI]: 0.68–2.19), 3.65% py (95%CI: 2.57–5.02) and 6.11% py (95%CI: 4.68–7.83) (**Table 2**).

The incidence of major thrombotic complications in the edoxaban group was 0.99% py (95%CI: 0.43–1.97), in the apixaban group was 2.16% py (95%CI: 0.45–6.30), and in the dabigatran group was 3.03% py (95%CI: 0.37–10.97), without statistically significant differences. The incidence of major hemorrhagic complications was 3.49% py (95%CI: 2.32–5.05) in the edoxaban cohort, 4.31% py (95%CI: 1.58–9.38) in the apixaban cohort, and 4.55% py (95%CI 0.94–13.27) in the dabigatran cohort, without statistically significant differences. The incidence rates of clinically relevant non-major thrombosis (CRNMT) were very similar between DOACs. The CRNMB incidence rate was 10.78% py in the apixaban group, 7.98% py in the edoxaban group, and 7.58% py in the dabigatran group.

The all-cause mortality was significantly higher in the apixaban group (10.78% py, 95%CI: 6.03–17.74, p<0.05) than the edoxaban group (5.12% py, 95%CI:3.67–6.94).

The rates of thrombotic and major bleeding events were very similar among different edoxaban doses, with higher all-cause mortality in the edoxaban 30 mg cohort (8.68% py vs 2.72% py, P<0.001).

**Table 2. Complications according to DOAC type.**

| | Overall | Edoxaban* | Edoxaban 30 mg | Edoxaban 60 mg | Apixaban* | Dabigatran* | Rivaroxaban* |
|---|---|---|---|---|---|---|---|
| **N** | 817 | 615 | 233 | 382 | 131 | 58 | 13 |
| **Total follow-up** (years) | 1015.16 | 801.62 | 322.76 | 478.86 | 139.18 | 65.94 | 8.42 |
| **Follow-up (months); med [$P_{25}$-$P_{75}$]** | 13.7 (7.5–22.8) | 15.0 (7.9–24.4) | 16.7 (8.4–25.1) | 13.6 (7.6–23.8) | 11.9 (6.7–18.8) | 11.5 (7.8–20.6) | 3.3 (1.5–13.5) |
| **Age; mean (SD)** | 77.7 (8.3) | 78 (7.8) | 81.4 (7.0) | 75.9 (7.6) | 77.7 (9.3) | 74.6 (10.1) | 76.2 (7.8) |
| **Major thrombotic complications (stroke/systemic embolism)** | | | | | | | |
| **$CHA_2DS_2$-VASc; mean (SD)** | 4.02 (1.57) | 3.97 (1.52) | 4.42 (1.49) | 3.69 (1.46) | 4.37 (1.78) | 3.78 (1.63) | 4.15 (1.46) |
| **Number of events** | 13 | 8 | 3 | 5 | 3 | 2 | 0 |
| **Incidence rate; % py (95%CI)** | 1.28 [0.68–2.19] | 0.99 [0.43–1.97] | 0.93 [0.19–2.72] | 1.04 [0.34–2.44] | 2.16 [0.45–6.30] | 3.03 [0.37–10.97] | 0.00 [0.00–43.81] |
| **Major hemorrhagic complications** | | | | | | | |
| **HAS-BLED; mean (SD)** | 2.17 (0.80) | 2.10 (0.76) | 2.14 (0.81) | 2.08 (0.73) | 2.45 (0.84) | 2.28 (1.04) | 2.23 (0.60) |
| **Number of events** | 37 | 28 | 11 | 17 | 6 | 3 | 0 |
| **Incidence rate; % py (95%CI)** | 3.65 [2.57–5.02] | 3.49 [2.32–5.05] | 3.41 [1.70–6.10] | 3.55 [2.07–5.68] | 4.31 [1.58–9.38] | 4.55 [0.94–13.27] | 0.00 [0.00–43.81] |
| **Composite (major thrombotic and hemorrhagic complications)** | | | | | | | |
| **Number of events** | 50 | 36 | 14 | 22 | 9 | 5 | 0 |
| **Incidence rate; % py (95%CI)** | 4.93 [3.67–6.49] | 4.49 [3.15–6.22] | 4.34 [2.37–7.28] | 4.59 [2.88–6.96] | 6.47 [2.96–12.28] | 7.58 [2.46–17.70] | 0.00 [0.00–43.81] |
| **Clinically relevant non-major thrombosis** | | | | | | | |
| **Number of events** | 10 | 8 | 3 | 5 | 2 | 0 | 0 |
| **Incidence rate; % py (95%CI)** | 0.99 [0.47–1.81] | 1.00 [0.43–1.97] | 0.93 [0.19–2.73] | 1.04 [0.34–2.44] | 1.44 [0.17–5.19] | 0.00 [0.00–5.59] | 0.00 [0.00–43.81] |
| **Clinically relevant non-major bleeding** | | | | | | | |
| **Number of events** | 85 | 64 | 24 | 40 | 15 | 5 | 0 |
| **Incidence rate; % py (95%CI)** | 8.37 [6.69–10.35] | 7.98 [6.15–10.20] | 7.44 [4.76–11.10] | 8.35 [5.97–11.38] | 10.78 [6.03–17.78] | 7.58 [2.46–17.70] | 0.00 [0.00–43.81] |
| **All-cause mortality** | | | | | | | |
| Number of events | 62 | 41 | 28 | 13 | 15 | 4 | 2 |
| Incidence rate; % py (95%CI) | 6.11 [4.68–7.83] | 5.12 [3.67–6.94] | 8.68 [5.77–12.54] | 2.72 [1.45–4.64] φ | 10.78 [6.03–17.78] † | 6.07 [1.65–15.53] | 23.75 [2.88–85.80] |

% py = per 100 patient-years

* Standard and reduced doses

† Statistically significant vs edoxaban group with P<0.05.

‡ Statistically significant vs edoxaban group with P<0.01.

φ Statistically significant vs edoxaban 30mg group with P<0.01.

**Table 3** shows the hazard ratios for thrombotic and hemorrhagic complications in the edoxaban cohort for each of the baseline and laboratory characteristics. CCI showed a statistically significant association with the first occurrence of a major thrombotic complication, major hemorrhagic complication and all bleeding complications (HR 1.61, 95%CI:1.23–2.09, p = 0.001; HR 1.27, 95%CI:1.01–1.60, p = 0.04; and HR 1.29, 95%CI:1.13–1.47, p<0.001 respectively, for each additional point of CCI). Lower hemoglobin was found to be a significant risk factor for both hemorrhagic (HR 0.97, 95%CI:0.95–0.99, p = 0.017) and all bleeding complications (HR 0.99, 95%CI:0.97–1.00, p = 0.047) and elevated anti-FXa activity trough (predose) level for all bleeding complications (HR 3.73, 95%CI:1.38–10.08, p = 0.009). ALT was associated with the first occurrence of a major thrombotic complication with a HR of 1.06 (95%CI:1.01–1.11, p = 0.022). Last but not least, a history of previous bleeding was shown

**Table 3. Clinical and laboratory characteristics related to thrombotic and hemorrhagic complications in the edoxaban cohort.**

| | Major thrombotic complications HR (95%CI) | P | Major hemorrhagic complications HR (95%CI) | P | CRNMB or major hemorrhagic complication HR (95%CI) | P |
|---|---|---|---|---|---|---|
| **Age** | 0.92 (0.84–1.01) | 0.086 | 1.04 (0.99–1.10) | 0.142 | 1.01 (0.98–1.04) | 0.579 |
| **Sex (male)** | 3.19 (0.64–15.81) | 0.156 | 1.18 (0.55–2.55) | 0.673 | 1.53 (0.96–2.44) | 0.072 |
| **Naive patients** | 1.53 (0.18–12.99) | 0.700 | 1.06 (0.32–3.57) | 0.924 | 0.73 (0.31–1.69) | 0.462 |
| **CHA$_2$DS$_2$-VASc** | 1.16 (0.74–1.82) | 0.512 | 1.01 (0.78–1.31) | 0.931 | 1.06 (0.91–1.24) | 0.446 |
| **HAS-BLED** | 1.66 (0.68–4.07) | 0.269 | 0.99 (0.58–1.69) | 0.962 | 1.10 (0.80–1.52) | 0.539 |
| **CCI**[*] | **1.61** (1.23–2.09) | **0.001** | **1.27** (1.01–1.60) | **0.040** | **1.29** (1.13–1.47) | **<0.001** |
| **High dose** | 1.19 (0.28–5.00) | 0.813 | 0.90 (0.41–1.96) | 0.791 | 1.03 (0.65–1.65) | 0.892 |
| **BMI**[***], kg/m$^2$ | 0.97 (0.83–1.14) | 0.740 | 1.04 (0.97–1.13) | 0.267 | 1.03 (0.98–1.08) | 0.250 |
| **Hemoglobin g/L** | 1.02 (0.98–1.07) | 0.287 | **0.97** (0.95–0.99) | **0.017** | **0.99** (0.97–1.00) | **0.047** |
| **Bilirubin** | 1.09 (0.93–1.27) | 0.303 | 1.02 (0.94–1.11) | 0.671 | 1.02 (0.97–1.08) | 0.433 |
| **AST** | 1.07 (0.97–1.17) | 0.194 | 1.00 (0.94–1.06) | 0.971 | 0.99 (0.95–1.03) | 0.559 |
| **ALT** | **1.06** (1.01–1.11) | **0.022** | 0.92 (0.85–1.00) | 0.061 | 0.97 (0.94–1.01) | 0.131 |
| **CrCl mean**[**] | 0.99 (0.96–1.03) | 0.655 | 0.99 (0.98–1.01) | 0.565 | 1.00 (0.99–1.01) | 0.601 |
| **Anti-FXa act. trough (UI/mL)** | 2.04 (0.04–102.3) | 0.722 | 4.13 (0.82–20.9) | 0.087 | **3.73** (1.38–10.08) | **0.009** |
| **Anti-FXa act. peak (UI/mL)** | 1.38 (0.29–6.62) | 0.687 | 1.77 (0.76–4.13) | 0.189 | 1.00 (0.60–1.67) | 0.995 |
| **Antiplatelets n, %** | 4.45 (0.90–22.09) | 0.068 | 1.82 (0.55–6.04) | 0.332 | 1.26 (0.55–2.91) | 0.585 |
| **History of previous stroke %** | 2.58 (0.52–12.78) | 0.246 | 0.68 (0.16–2.87) | 0.597 | 1.01 (0.49–2.10) | 0.980 |
| **History of previous bleeding %** | 1.94 (0.24–15.91) | 0.538 | 1.54 (0.46–5.16) | 0.482 | **2.47** (1.32–4.61) | **0.005** |

[*] CCI: Charlson Comorbidity Index; its HR value refers to each point increase in the index.

[**] CrCl: creatinine clearance from Cockcroft-Gault formula (mL/min).

[***] Body mass index

to be a risk factor for all bleeding complications, with a HR of 2.47 (95%CI:1.32–4.61, p = 0.005).

## Subgroup analyses

**S1 Table in S1 File** describes the characteristics of the 615-patient edoxaban population stratified by age (≤74 years, 75–84 years and ≥85 years) and by sex. When analyzed by age, the proportion of male patients decreased from 57.4% in the ≤74 year-old group to 43.9% in the ≥85 year-old group. The thrombotic risk as measured with CHA$_2$DS$_2$-VASc increased (3.03 to 4.65 points), as did the history of previous ischemic stroke (8.7% to 18.2%). The bleeding risk as measured with HAS-BLED was highest in the oldest group (1.95 in ≤74-year-olds to 2.21 points in ≥85-year-olds). Mean CrCl decreased with age (77.2 to 47.9mg/mL). The proportion of naive patients, CCI and history of previous major bleeding appeared similar across all ages.

The incidence of major thrombotic complications was similar across age groups, but the incidence of major hemorrhagic complications increased notably with age (1.69%, 3.91% and 5.13% py), with only the difference between the youngest and the oldest cohorts reaching statistical significance. The all-cause mortality rate was markedly higher in patients aged ≥85 years: 11.39% py vs 3.39% py in those aged 75–84 years (**S2 Table in S1 File**).

When analyzed by sex (S1 and S2 Tables in **S1** File), female patients were older (79.2 y vs 76.8 y), had worse mean CHA$_2$DS$_2$-VASc score (4.40 vs 3.55) and better HAS-BLED score (2.02 vs 2.18) and CCI (1.17 vs 1.73). 12 patients with GFR <30mL/min were female vs one

male. Male patients used more concomitant antiplatelet therapy (10.8% vs 2.7% in women). There were no statistically significant differences in outcomes depending on sex.

**DOAC results in anticoagulant-naïve patients only.** To ensure that the trends in complications in our study were not related to previous vitamin K antagonist therapy, we performed an analysis in the anticoagulant-naive subgroup of 185 patients (**S3 Table in S1 File**). In these naive patients, the incidence of major thrombotic and bleeding complications was 2.40% and 3.00% py respectively. This gave a composite of 5.41% py with a mortality of 8.41% py.

## Discussion

We have presented the clinical outcomes of an unselected cohort of patients who started treatment with DOACs and followed a systematic care protocol.

The absolute incidence of major complications is the best way to measure the efficacy and safety of any anticoagulant drug in the real world.

To better understand the observed differences in clinical outcomes, we suggest always keeping in mind two critical variables: the mean age and the mean $CHA_2DS_2$-VASc of each study. Unfortunately, we usually lack information on the clinical models of DOAC use, but it is reasonable to suppose that, in the large-scale registers, those models must be very heterogeneous [5–12].

In our overall cohort, there was an incidence of thrombotic complications of 1.28% py; of major hemorrhagic complications, 3.65% py; and of total complications, 4.93% py. The mortality incidence was 6.11% py. We have used these figures to compare our results with other studies in **Tables 4 and 5**.

The selection of these comparative examples does not come from a systematic review. The criteria to select the publications are as follows: first, pivotal studies of the 4 DOACs currently used in the real world; second, real-life results from different geographical areas (Europe, America, and Asia); then, registries with a very high number of patients or registries with a methodology that suggests highly reliable results regardless the number of patients enrolled (in the subjective opinion of the authors). In addition, we included frequently cited publications. We believe that our selection is representative of the literature on DOACs in the real world.

### Edoxaban results

Our sample of 615 patients on edoxaban who were followed up for 15.0 months allows us to make reliable estimates of the incidence of major complications. Our population is older and with a worse $CHA_2DS_2$-VASc score than most previous studies (Tables 4 and 5). We observed an incidence of 4.49% py of total major complications, with a predominance of major bleeding (3.49% py). When the analysis was split by edoxaban dose, we did not observe differences in incidence of complications, but there was a higher mortality in the 30 mg group, probably due to the clinical characteristics of that group.

Focusing on the age subgroup analysis (S1 and S2 Tables in **S1** File), the increased complexity of patients with increasing age was reflected in the number of major hemorrhagic complications; meanwhile, the incidence of thromboembolic events remained stable. This is important to keep in mind while treating elderly patients with DOACs. There were no differences in terms of outcomes when analyzed by sex.

Regarding demographic characteristics related to complications, we found that every increasing point of the Charlson Comorbidity Index [38] represented an increased risk of a first occurrence of a major thrombotic or hemorrhagic complication and of all bleeding complications. Risk factors related to bleeding were low hemoglobin, anti-FXa activity trough (pre-dose) level and history of previous bleeding, but not age.

**Table 4. Main outcomes with DOACs from studies in real practice or RCTs, reporting a lower or similar incidence of major complications than MACACOD.**

| Author | Ref | Country | Design[1] | Drug | N° of patients | Age in years (mean) | CHA$_2$DS$_2$-VASc (mean) | Incidence of major complications(% patient-year) | | | |
|---|---|---|---|---|---|---|---|---|---|---|---|
| | | | | | | | | Thromb[2] | Hemorr | Total* | Death |
| Connolly (RELY) | [1] | Multinational | RCT | Dab 150 | 6,076 | 71.5 | 2.2 [&] | 1.1 | 3.1 | 4.2* | 3.6 |
| | | | | Dab 110 | 6,015 | 71.4 | 2.1 [&] | 1.5 | 2.7 | 4.2* | 3.7 |
| Patel (ROCKET) | [2] | Multinational | RCT | Riv | 7,131 | 73.0 | 3.5 [&] | 1.7 | 3.6 | 5.3* | 1.9 |
| Granger (ARISTOTLE) | [3] | Multinational | RCT | Apix | 9,120 | 70.0 | 2.1 [&] | 1.3 | 2.1 | 3.2 | 3.5 |
| Giugliano (ENGAGE) | [4] | Multinational | RCT | Edox 60 | 7,035 | 72.0 | 2.8 [&] | 1.2 | 2.7 | 3.9* | 4.0 |
| | | | | Edox 30 | 7,034 | 72.0 | 2.8 [&] | 1.6 | 1.6 | 3.2* | 3.8 |
| Graham | [20] | USA | RC | Dab | 67,207 | 77.0 ** | 2.1[&$] | 1.1 | 4.3 | 5.4* | 3.3 |
| Larsen | [41] | Denmark Nationwide | RC | Dab | 12,701 | 67.6 | 2.2 | 1.8 | 2.0 | 3.8* | 2.4 |
| | | | | Riv | 7,192 | 71.8 | 2.8 | 2.3 | 3.6 | 5.9* | 6.7 |
| | | | | Apix | 6,349 | 71.3 | 2.8 | 4.1 | 2.7 | 6.8* | 4.8 |
| Graham | [5] | USA | RC | Dab | 52,240 | 75.8** | 2.3[&$] | 1.0 | 3.1 | 4.1* | 2.2 |
| | | | | Riv | 66,651 | 75.6** | 2.3[&$] | 0.8 | 4.5 | 5.3* | 2.5 |
| Carmo | [21] | Multinational | MA | Dab | 210,279 | 66.0–79.5 | 1.2–4.8 | 1.6 | 3.9 | 5.5* | 3.6 |
| Camm | [5] | Multinational | PC | Riv | 6,784 | 71.5 | 3.4 | 0.8 | 2.1 | 2.9* | 1.9 |
| Mentias | [22] | USA Medicare | RC | Dab | 21,979 | 75.8 | 4.3 | 1.5 | 3.4 | 4.9* | 2.6 |
| | | | | Riv | 23,177 | 75.7 | 4.3 | 1.4 | 4.7 | 6.1* | 3.1 |
| Chan | [44] | Taiwan Nationwide | RC | Dab | 20,079 | 75.0 | 3.7 | 2.7 | 2.0 | 4.7* | 5.0 |
| | | | | Riv | 27,777 | 75.0 | 3.8 | 2.8 | 2.1 | 4.9* | 6.0 |
| | | | | Apix | 5,843 | 76.0 | 3.9 | 2.3 | 1.5 | 3.8* | 7.2 |
| Gupta | [45] | USA DOD registry | RC | Dab | 4,129 | 73.0 | 3.6 | 1.0 | 3.6 | 4.6* | NR |
| | | | | Riv | 11,284 | 75.3 | 4.0 | 1.3 | 4.4 | 5.7* | NR |
| | | | | Apix | 11,284 | 75.3 | 4.0 | 1.0 | 2.9 | 3.9* | NR |
| Vinogradova | [13] | United Kingdom | PC | Dab | 4,534 | 74.7 | NR | 1.7 | 2.2 | 3.9* | 4.3 |
| | | | | Riv | 13,597 | 75.8 | NR | 1.6 | 2.6 | 4.2* | 5.5 |
| | | | | Apix | 9,199 | 76.5 | NR | 1.8 | 1.5 | 3.3* | 5.3 |
| Lip | [46] | USA | RC | Dab | 36,990 | 73.2 | 3.5 | 1.3 | 3.6 | 4.9* | NR |
| | | | | Riv | 125,068 | 75.6 | 3.8 | 1.2 | 5.8 | 7.0* | NR |
| | | | | Apix | 100,977 | 76.1 | 3.9 | 1.3 | 3.6 | 4.9* | NR |
| Ujeyl | [18] | Germany | RC | Dab | 23,654 | 75.4 | 4.0 [$] | 1.8 | 3.4 | 5.2* | 7.2 |
| | | | | Riv | 59,449 | 75.4 | 4.0 [$] | 1.5 | 4.1 | 5.6* | 8.8 |
| | | | | Apix | 4,894 | 75.4 | 4.0 [$] | 1.9 | 2.6 | 4.5* | 9.5 |
| Gupta | [47] | USA | RC | Dab | 3,691 | 74.0 | 3.7 | 1.0 | 3.7 | 4.7* | NR |
| | | | | Riv | 8,226 | 76.5 | 4.1 | 1.5 | 5.0 | 6.5* | NR |
| | | | | Apix | 7,607 | 76.5 | 4.2 | 0.9 | 3.1 | 4.0* | NR |
| Lee | [6] | Korea Nationwide | RC | Dab | 17,745 | 70.8 | 3.5 | 2.3 | 1.5 | 3.8* | NR |
| | | | | Riv | 35,965 | 72.0 | 3.6 | 2.4 | 2.1 | 4.5* | NR |
| | | | | Apix | 22,177 | 72.7 | 3.8 | 2.3 | 1.8 | 4.1* | NR |
| | | | | Edox | 15,496 | 71.7 | 3.6 | 2.0 | 1.6 | 3.6* | NR |
| Amin | [48] | USA | RC | Dab | 18,131 | 77.1 | 4.4 | 1.3 | 4.1 | 5.4* | NR |
| | | | | Riv | 55,359 | 77.9 | 4.5 | 1.0 | 6.1 | 7.1* | NR |
| | | | | Apix | 37,525 | 78.4 | 4.6 | 1.0 | 3.7 | 4.7* | NR |
| Graham | [7] | USA | RC | Dab | 86,293 | 75.5 | 3.6 [$] | 0.9 | 3.0 | 3.9* | 2.1 |
| | | | | Riv | 106,369 | 74.9 | 3.6 [$] | 0.8 | 4.1 | 4.9* | 2.4 |
| | | | | Apix | 73,039 | 75.2 | 3.6 [$] | 0.9 | 1.9 | 2.8* | 2.1 |

(*Continued*)

**Table 4.** (Continued)

| Author | Ref | Country | Design[1] | Drug | N° of patients | Age in years (mean) | $CHA_2DS_2$-VASc (mean) | Incidence of major complications(% patient-year) | | | |
| --- | --- | --- | --- | --- | --- | --- | --- | --- | --- | --- | --- |
| | | | | | | | | Thromb[2] | Hemorr | Total* | Death |
| Villines | [8] | USA | RC | Dab | 12,763 | 70.9 | 3.1 | 0.5 | 1.8 | 2.3* | 1.8 |
| | | | | Riv | 12,763 | 70.9 | 3.1 | 0.7 | 2.2 | 2.9* | 1.6 |
| | | | | Apix | 4,802 | 70.2 | 3.0 | 0.4 | 1.2 | 1.6* | 1.2 |
| Kohsaka | [9] | Japan | RC | Dab | 6,925 | 72.5 | 3.4 | 1.6 | 1.4 | 3.0* | NR |
| | | | | Riv | 16,564 | 74.2 | 3.6 | 1.6 | 1.6 | 3.2* | NR |
| | | | | Apix | 22,336 | 77.0 | 3.9 | 1.7 | 1.8 | 3.5* | NR |
| | | | | Edox | 12,262 | 76.3 | 3.8 | 1.9 | 1.8 | 3.7* | NR |
| Marietta | [14] | Italy | PC | D+R+A | 2,178 | 75.4 | 3.4 | 0.8 | 1.9 | 2.7* | 1.7 |
| Jansson | [42] | Sweden | RC | Dab | 6,453 | 72.3 | 2.6 | 1.1 | 2.3 | 3.4* | 3.0 |
| | | | | Riv | 7,897 | 73.5 | 3.1 | 1.6 | 4.2 | 5.8* | 5.8 |
| | | | | Apix | 11,493 | 73.5 | 3.1 | 1.4 | 2.8 | 4.2* | 6.1 |
| Rutherford | [43] | Norway Nationwide | RC | Dab | 10,431 | 70,7 | 3.0 | 1.8 | 1.4 | 3.2* | NR |
| | | | | Riv | 13,700 | 71.8 | 3.1 | 2.2 | 2.0 | 4.2* | NR |
| | | | | Apix | 28,363 | 71.7 | 3.1 | 2.6 | 1.6 | 4.2* | NR |
| Huybrechts | [19] | USA Hospitalized | PC | Dab | 29,448 | 67.8 | 3.0 | 0.9 | 4.0 | 4.9 | 0.8 [ψ] |
| | | | | Riv | 35,520 | NR | NR | 1.0 | 6.7 | 7.7 | NR |
| | | | | Apix | 19,588 | NR | NR | 0.8 | 4.0 | 4.8 | NR |
| Ray | [10] | USA Medicare | RC | Apix | 353,879 | 77.0 | 4.3 | 0.8 | 2.6 | 3.4* | 4.5 |
| | | | | Riv | 227,572 | 77.0 | 4.3 | 0.8 | 4.3 | 5.1* | 3.9 |
| De Groot | [12] | Multinational | PC | Edox 60 | 9,991 | 71.8 | 2.9 | 0.8 | 0.9 | 1.7* | 2.4 |
| | | | | Edox 30 | 3,101 | 79.5 | 3.8 | 1.0 | 1.6 | 2.6* | 7.2 |
| Camm | [49] | Multinational | PC | Dab | 2,090 | 72.0** | 4.0** | 0.9 | 0.7 | 1.6* | 3.3 |
| | | | | R + A | 6,790 | 75.0** | 4.0** | 0.8 | 1.0 | 1.8* | 3.7 |
| MACACOD | | Spain | PC | All DOACs | 817 | 77.7 | 4.02 | 1.28 | 3.65 | 4.93 | 6.11 |
| | | | | Edox | 615 | 78 | 3.97 | 0.99 | 3.49 | 4.49 | 5.12 |

1 Types of study: RCT, randomized controlled trial. RC: retrospective cohort. PC: prospective cohort. MA: Meta-analysis

NR: Not reported. DOD: US Department of Defense. Dab: Dabigatran. Riv: Rivaroxaban. Apix: Apixaban. Edox: Edoxaban. D+R+A, dabigatran, rivaroxaban and apixaban.

2. Thrombotic means ischemic stroke or systemic embolism.

*The incidence of total complications was estimated (when not explicitly reported) by the sum of thromboembolic + hemorrhagic complications

** Median

& $CHADS_2$

$ Estimated by the authors

ψ In-hospital

In **Table 6** we compare our results with other reports on edoxaban. Our findings are slightly worse than those of the pivotal ENGAGE RCT [4], which had a population with a mean age six years younger, but with a similar $CHA_2DS_2$-VASc score (considering that $CHADS_2$ score is approximately one point lower than $CHA_2DS_2$-VASc).

Kohsaka et al [9] and Lee et al [6] reported slightly better results than our study but included only complications in hospitalized patients [9] or younger patients with lower $CHA_2DS_2$-VASc scores [6]. In contrast, the results from the ETNA registry [12], which were very good, even better than the pivotal study, could have been influenced by some of the

**Table 5. Main outcomes with DOACs from studies reporting a similar or higher incidence of major complications than MACACOD.**

| Author | Ref | Country | Design[1] | Drug | Number of patients | Age in years (mean) | CHA$_2$DS$_2$-VASc (mean) | Incidence of major complications (% patient-year) | | | |
| --- | --- | --- | --- | --- | --- | --- | --- | --- | --- | --- | --- |
| | | | | | | | | Thromb[2] | Hemorr | Total* | Death |
| Van Mieghem | [26] | Multinational | RCT | Edox | 713 | 82.1 | 4.5 | 2.3 | 9.7 | 12.0* | 7.8 |
| Nielsen φ | [27] | Denmark Nationwide | RC | Dab | 8,875 | 79.9 | 3.8 | 3.3 | 3.0 | 6.3* | 11.0 |
| | | | | Riv | 3,476 | 77.9 | 3.6 | 2.9 | 4.4 | 7.3* | 18.2 |
| | | | | Apix | 4,400 | 83.9 | 4.3 | 5.6 | 3.8 | 9.4* | 23.8 |
| Lee | [28] | Korea | RC | Edox 60 | 1,835 | 66.7 | 2.8 | 2.3 | 3.2 | 5.5* | 3.2 |
| | | | | Edox 30 | 2,371 | 73.8 | 3.6 | 4.1 | 6.4 | 10.5* | 8.6 |
| Yu | [29] | Korea Nationwide | RC | Edox 60 | 2,840 | 68.2 | 4.2 | 4.3 | 3.1 | 7.4* | 2.7 |
| | | | | Edox 30 | 3,016 | 72.8 | 4.9 | 6.4 | 5.5 | 11.9* | 7.3 |
| Mueller | [52] | Scotland | RC | Dab | 1,112 | 71.1 | 2.5 | 1.8 | 4.1 | 5.9* | 5.2 |
| | | | | Riv | 7,265 | 74.8 | 3.0 | 1.8 | 6.3 | 8.1* | 9.4 |
| | | | | Apix | 6,200 | 73.7 | 2.9 | 1.8 | 4.7 | 6.5* | 7.5 |
| Bang | [15] | Korea Nationwide | RC | Dab | 11,414 | 72.2 | 4.6 | 7.1 | 7.8 | 14.9* | NR |
| | | | | Riv | 17,779 | 73.3 | 4.6 | 7.1 | 9.4 | 16.5* | NR |
| | | | | Apix | 10,548 | 73.9 | 4.8 | 8.0 | 8.3 | 16.3* | NR |
| Nielsen-Kudsk | [17] | Denmark | RC | All DOACs | 1,184 | 75.1 | 4.3 | 1.9 | 10.0 | 11.9* | 15.3 |
| Chen # | [16] | Netherlands Nationwide | RC | AF patients 47% DOACS 25% VKA 28% no OAT | 342,209 | NR | NR | 6.7 | 4.1 | 10.8* | 20.1 |
| Buderi | [50] | USA | RC | All DOACs | 61,214 | 72.2 | 4.0 | NR | 19.0 ∞ | NR | NR |
| Nielsen | [30] | Denmark Nationwide | RC | Edox 60 | 1,642 | 73.0 | 3.2 | 1.6 | 3.9 | 5.5* | 6.3 |
| | | | | Edox 30 | 643 | 80.5 | 4.2 | 2.1 | 3.9 | 6.0* | 16.5 |
| Paschke | [40] | Germany Nationwide | RC | Dab | 55,131 | 74.4 | 4.15 | 4.9 | NR | NR | 6.3 |
| | | | | Riv | 234,802 | 75.0 | 4.21 | 3.1 | NR | NR | 7.5 |
| | | | | Apix | 133,970 | 76.6 | 4.44 | 4.2 | NR | NR | 7.4 |
| | | | | Edox | 14,666 | 75.5 | 4.28 | 1.7 | NR | NR | 3.1 |
| MACACOD | | Spain | PC | All DOACs | 817 | 77.7 | 4.02 | 1.28 | 3.65 | 4.93 | 6.11 |
| | | | | Edox | 615 | 78 | 3.97 | 0.99 | 3.49 | 4.49 | 5.12 |
| | | | | Edox 60 | 382 | 75.9 | 3.69 | 1.04 | 3.55 | 4.59 | 2.72 |
| | | | | Edox 30 | 233 | 81.4 | 4.42 | 0.93 | 3.41 | 4.34 | 8.68 |

Dab: Dabigatran; Riv: Rivaroxaban; Apix: Apixaban; Edox: Edoxaban.

1. Types of study: RCT, randomized controlled trial. RC, retrospective cohort. PC, prospective cohort.

2. Thrombotic means ischemic stroke or systemic embolism.

*The incidence of total complications was estimated by the sum of thromboembolic + hemorrhagic complications

φ All patients received the reduced dose of their respective DOAC. Incidence rates are crude and calculated at 2.5 years of follow-up.

# The data correspond to the year 2017. As they are sets of the entire population with AF, they do not provide exact values for DOACS.

∞ This high incidence probably includes CRNMB

problems mentioned above. The edoxaban results from Lee [6], Yu [29], Nielsen [30], Marston [39], and Paschke [40] are slightly worse than ours. The Korean studies [6, 29] found a higher incidence of complications with edoxaban 30 mg than edoxaban 60 mg, but this was not found in the other studies.

**Table 4** lists multiple studies reporting a lower incidence of major complications [1–10, 12–14, 18–22, 41–49], and **Table 5** lists those that report a higher incidence. To be conservative, we used as a threshold the incidences observed in our total sample of all four DOACs. By

**Table 6. Main outcomes with edoxaban from studies in real world or RCTs, in comparison with results from MACACOD, stratified by age.**

| Author | Ref | Country | Design[1] | Drug | Number of patients | Age in years (mean) | CHA$_2$DS$_2$-VASc (mean) | Incidence of major complications(% patient-year) | | | |
|---|---|---|---|---|---|---|---|---|---|---|---|
| | | | | | | | | Thromb[2] | Hemorr | Total* | Death |
| Giugliano (ENGAGE) | [4] | Multinational | RCT | Edox 60 | 7,035 | 72.0 | 2.8 [&] | 1.2 | 2.7 | 3.9* | 4.0 |
| | | | | Edox 30 | 7,034 | 72.0 | 2.8 [&] | 1.6 | 1.6 | 3.2* | 3.8 |
| Van Mieghem Nej | [26] | Multinational | RCT | Edox | 713 | 82.1 | 4.5 | 2.3 | 9.7 | 12.0* | 7.8 |
| Lee | [28] | Korea | RC | Edox 60 | 1,835 | 66.7 | 2.8 | 2.3 | 3.2 | 5.5* | 3.2 |
| | | | | Edox 30 | 2,371 | 73.8 | 3.6 | 4.1 | 6.4 | 10.5* | 8.6 |
| Yu | [29] | Korea Nationwide | RC | Edox 60 | 2,840 | 68.2 | 4.2 | 4.3 | 3.1 | 7.4* | 2.7 |
| | | | | Edox 30 | 3,016 | 72.8 | 4.9 | 6.4 | 5.5 | 11.9* | 7.3 |
| Lee | [6] | Korea | RC | Edox | 15,496 | 71.7 | 3.6 | 2.0 | 1.6 | 3.6* | NR |
| Kohsaka | [9] | Japan | RC | Edox | 12,262 | 76.3 | 3.8 | 1.9 | 1.8 | 3.7* | NR |
| De Groot | [12] | Multinational | PC | Edox 60 | 9,991 | 71.8 | 2.9 | 0.8 | 0.9 | 1.7* | 2.4 |
| | | | | Edox 30 | 3,101 | 79.5 | 3.8 | 1.0 | 1.6 | 2.6* | 7.2 |
| Nielsen | [30] | Denmark Nationwide | RC | Edox 60 | 1,642 | 73.0 | 3.2 | 1.6 | 3.9 | 5.5* | 6.3 |
| | | | | Edox 30 | 643 | 80.5 | 4.2 | 2.1 | 3.9 | 6.0* | 16.5 |
| Paschke | [40] | Germany Nationwide | RC | Edox | 14.666 | 75.5 | 4.3 | 1.7 | NR | NR | 3.1 |
| MACACOD | | Spain | PC | Edox | 615 | 78 | 3.97 | 0.99 | 3.49 | 4.49 | 5.12 |
| | | | | Edox 60 | 382 | 75.9 | 3.69 | 1.04 | 3.55 | 4.59 | 2.72 |
| | | | | Edox 30 | 233 | 81.4 | 4.42 | 0.93 | 3.41 | 4.34 | 8.68 |
| | | | | Edox ≤ 74y | 195 | 68.9 | 3.03 | 1.69 | 1.69 | 3.39 | 2.96 |
| | | | | Edox 75-84y | 288 | 79.6 | 4.29 | 0.52 | 3.91 | 4.43 | 3.39 |
| | | | | Edox ≥ 85y | 132 | 87.8 | 4.65 | 1.14 | 5.13 | 6.27 | 11.39 |

Edox: Edoxaban.

1. Types of study: RCT, randomized controlled trial. RC, retrospective cohort. PC, prospective cohort

2. Thrombotic means ischemic stroke or systemic embolism.

*The incidence of total complications was estimated by the sum of thromboembolic + hemorrhagic complications

[&] CHADS$_2$

so doing, our results compare well with all the previous results, since those with lower incidences are generally shown to be in younger populations, with less frailty or comorbidity. Some of the studies with worse outcomes than ours are also in younger populations, although in general they have similar or higher CHA$_2$DS$_2$-VASc scores [15, 28–30, 40, 50].

Many of the studies summarized in **Table 4** have remarkably good results. In their critical evaluation, some points that raise doubts about credibility should be considered. First, it is difficult to obtain similar or even lower incidences of serious complications in clinical practice [5–14, 22] than in RCTs [1–4], as already pointed out in the study by de Vries et al [51], for the simple reason that patients in real life are older and have more comorbidity. However, the data they offer on mortality (repeatedly very low) do not align with the expected mortality of 16% py in the first year [23, 24] and 8% py in the following years after AF diagnosis [25]. This 8% refers to a population with a mean age of 70 years and a CHA$_2$DS$_2$-VASc score of 2.8, that is, with a lower risk than the populations in the studies we criticize.

The data in **Table 5** seem more consistent with reality and, if true, are certainly worrying due to the high incidence of serious complications. The total complications exceed 10% py in almost all the studies. The best incidences in this table come from nationwide studies from

Denmark [27, 30] and a Scottish study [52] and are between 5.5 and 9.4% py, perhaps due to their better health care models. Note that the all-cause mortalities in these studies in are very close to what would be expected [23–25].

However, these high incidences of severe complications with DOACs, are lower than those observed with VKA therapy in real world [15, 28, 29, 39, 53]. In the Marston et al study [39], the composite incidence of major thrombotic and hemorrhagic events was 8.8% py with dabigatran, 6.83% py with apixaban, 7.18% py with rivaroxaban and 5.40% py with edoxaban but 11.82% py with VKA treatment.

Perhaps the key to these inconsistencies is found in the article by Bang et al [15], where a huge difference (between half and a third) was observed in the crude incidence rates of major complications when the diagnoses were limited to inpatient claims. Or in the works by Koshaka et al [9], Huybrechts et al [19], and Ujeyl et al [18], which were limited to complications that required hospitalization or deaths that occurred in hospital.

The outcomes of the MACACOD project, may indicate that the specialized management of DOACs helps minimize complications to acceptable levels, but there is undoubtedly still room for improvement. We observed a favorable trend for edoxaban, compared to dabigatran and apixaban. This trend is in line with recent nationwide publications, three of them technically well-designed with propensity-score matching comparisons in Korea [6] and Germany [39, 40], and others in Denmark [27, 30], with lower incidences of major complications with edoxaban than with dabigatran, rivaroxaban, and apixaban.

## A critical appraisal of DOACs in daily practice

Marietta et al [54] warned about the difficulty of translating the results of trials into real life because, in an everyday care setting, patients have more comorbidities, are treated without a strict protocol, and are under less intensive surveillance than in an RCT. In addition, we must consider the concerning figures on non-adherence and non-persistence in the treatment with DOACs, which will inevitably worsen clinical results in the real world [55, 56].

One possible option to overcome such difficulties is the management by specialized clinics, where the clinical outcomes are expected to be as good as those presented by some reports [57] and high-quality registries [41, 42]. Our results are as good as those obtained in reliable studies such as those by Rutherford et al [43] in Norway with a population less vulnerable than ours, and even better than those reported by Nielsen et al [27, 30] in Denmark with populations very similar to ours.

This literature of DOACs suggests that the real situation is worse than is generally believed. If the figures for serious complications are like those reported by many of the studies in Table 5, then, "Houston, we have a problem..." The following inconsistency highlights this: the original $CHA_2DS_2$-VASc risk scale estimates the risk of thromboembolic events at 4% py for patients with a score of 3–6 [32]. A subsequent adjustment placed the risk between 4% and 6.7% py for scores of 4–5 [58]. A rate of total serious complications higher than 10% (obviously including hemorrhages) leads to an absurd situation, when what we are trying to prevent has an approximate risk of 4–6%. Our results, together with others, indicate the need for more highly-specialized clinical care when prescribing any DOAC.

## Strengths and limitations

The relatively small size of our study is both a limitation and a strength. It can be considered a strength because it made possible to guarantee the registration of incidents through personalized follow-up of patients and all their clinical records, both in and out of the hospital. The single-center nature of the study also guarantees a uniform clinical protocol. The main limitation

is the relatively small sample size, which mainly affected apixaban, dabigatran, and rivaroxaban, for which we could not make reliable estimates. Another limitation is, due to the nature of the prescription of DOACs in Spain, that they are used as a second-line therapy after VKAs. This translates into a low percentage of naive patients and, consequently, can bias comparisons with other studies.

## Conclusions

Although the vast majority of studies on clinical outcomes with DOACs in real life present low or very low incidences of serious complications, these may be underestimated. The complication rates in other recent credible studies are much higher and unacceptable.

The clinical management model proposed by MACACOD seems to be a good way to reduce complications. We observed an incidence of serious complications of 4.93% py, in which severe bleeding predominated (3.65% py). In our results, it seems that the edoxaban group performs better than other DOACs, as shown in some propensity-score real-life nationwide data from Lee [6], Maston [39], and Paschke [40]. Otherwise, our DOAC groups are not equal in terms of ages or basal risks, so these differences in co-variates could explain the observed differences in outcomes. More RCT or at least patient-level meta-analysis are necessary to evaluate the differences in safety and effectiveness between DOACs.

Contrary to the simplistic view of the ease of application and management of DOACs, greater specialization is necessary, as well as a better application of international guidelines and especially patient education regarding their anticoagulant treatment.

## Supporting information

**S1 Checklist. STROBE statement—checklist of items that should be included in reports of observational studies.**
(DOC)

**S1 File.**
(PDF)

## Author Contributions

**Conceptualization:** René Acosta-Isaac, Sergi Mojal, Blanca Jiménez, Melania Plaza, Juan Carlos Souto.

**Data curation:** René Acosta-Isaac, Sergi Mojal, Mariana Corrochano, Melania Plaza, Juan Carlos Souto.

**Funding acquisition:** Juan Carlos Souto.

**Investigation:** Carla Moret.

**Methodology:** Sergi Mojal.

**Project administration:** Juan Carlos Souto.

**Software:** Sergi Mojal.

**Supervision:** Juan Carlos Souto.

**Writing – original draft:** Carla Moret.

**Writing – review & editing:** Carla Moret, René Acosta-Isaac, Sergi Mojal, Mariana Corrochano, Blanca Jiménez, Melania Plaza, Juan Carlos Souto.

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
