## [Decision Letter · Decision Letter 0]

19 Sep 2022

PONE-D-22-21794Clinical outcomes in patients with atrial fibrillation treated with DOACs in a specialized anticoagulation center: Critical appraisal of real-world dataPLOS ONE

Dear Dr. Moret Puig,

Thank you for submitting your manuscript to PLOS ONE. After careful consideration, we feel that it has merit but does not fully meet PLOS ONE’s publication criteria as it currently stands. Therefore, we invite you to submit a revised version of the manuscript that addresses the points raised during the review process.

We look forward to receiving your revised manuscript.

Kind regards,

Giulio Francesco Romiti

Academic Editor

PLOS ONE

Journal Requirements:

"This study was funded with an unconditional grant from Daiichi-Sankyo Spain"

"The Hemostasis and Thrombosis Unit and the IIB-Sant Pau have received funding from Daiichi-Sankyo to develop and maintain the MACACOD registry (Clinical Application Model of Direct Oral AntiCoagulants). The funders had no role in study design, data collection and analysis, decision to publish, or preparation of the manuscript."

"JCS has received honoraria or financial support for travel, accommodation, or expenses from Laboratorios Rovi, Leo Pharma, Baxter, Sanofi, Boehringer Ingelheim, Pfizer, Bristol Myers Squibb, Roche, Daiichi-Sankyo, Stago Laboratories and performs an advisory role for Devicare. 

CM, RA, SM, MP, MC, BJ have declared that no competing interests exist."

We note that you received funding from a commercial source: Laboratorios Rovi, Leo Pharma, Baxter, Sanofi, Boehringer Ingelheim, Pfizer, Bristol Myers Squibb, Roche, Daiichi-Sankyo, Stago Laboratories

Within this Competing Interests Statement, please confirm that this does not alter your adherence to all PLOS ONE policies on sharing data and materials by including the following statement: ""This does not alter our adherence to PLOS ONE policies on sharing data and materials.” (as detailed online in our guide for authors http://journals.plos.org/plosone/s/competing-interests).  If there are restrictions on sharing of data and/or materials, please state these. Please note that we cannot proceed with consideration of your article until this information has been declared. 

5. Please include a caption for Figure 1.

Additional Editor Comments:

Please address carefully the reviewers' comments.

Reviewers' comments:

Reviewer's Responses to Questions

**Comments to the Author**

1. Is the manuscript technically sound, and do the data support the conclusions?

Reviewer #1: Yes

Reviewer #2: Partly

2. Has the statistical analysis been performed appropriately and rigorously? 

Reviewer #1: Yes

Reviewer #2: Yes

3. Have the authors made all data underlying the findings in their manuscript fully available?

Reviewer #1: Yes

Reviewer #2: Yes

4. Is the manuscript presented in an intelligible fashion and written in standard English?

Reviewer #1: Yes

Reviewer #2: Yes

5. Review Comments to the Author

Reviewer #1: I think that is a quite well-cnducted study.

I think that it is important in the conclusion part to explicit in which dimensions of the outcome edoxaban perform better than other DOAC, also because this is a founded study and I think that it is more ethical.

The novelty of the paper is very low, but real-word data in DOACs are always appreciated.

Reviewer #2: Authors present a prospective, observational, single center study enrolling over 600 patients with atrial fibrillation (MACACOD study) addressing the issue of efficacy and safety of DOAC in a real world setting. Below my major and minor concerns:

MAJOR

- The paper begins as an observational prospective single center study (title, abstract, materials and methods, results) and ends as a non-systematic review of the literature (Discussion). I think it goes beyond the scope of an observational study to comment on the whole literature present. Furthermore, no mention has been made on how some studies have been selected and some others have been discarded. Discussion section should focus instead on interpretation of the interesting results that you presented in the paper, trying to pinpoint what is relevant to the reader and why, and underlining the eventual limitation and possible bias. You might of course compare your results to those of other studies, but Table 4,5 and 6 go beyond this scope.

- Edoxaban and Apixaban population in this study are very different (percentage of naïve patients, antiplatelets drugs, previous bleeding, previous stroke). When presenting non pre-specified, subgroup analysis comparing two very different populations care must be taken not to over-emphasize differences that could easily be explained by the presence of a bias. Such results could be at most “hypothesis generating” and certainly can not support that “In our experience, edoxaban appears to perform better than other DOACs”. In order to state this, you need a RCT or at least a patient level meta-analysis of the available RCTs.

MINOR

- In the section “Materials and methods - Blood tests” authors described meticulously how FXa activity was monitored; by the way, no mention is present in the results section about such an experiment

- Results are often described as statistically significant without reporting confidence intervals and p-value. Even though these are present in the tables, they should be present also in the results section.

6. PLOS authors have the option to publish the peer review history of their article (what does this mean?). If published, this will include your full peer review and any attached files.

Reviewer #1: **Yes: **NICCOLO BONINI

Reviewer #2: No

---

## [Author Response · Author response to Decision Letter 0]

3 Nov 2022

Responses to the reviewers:

Reviewer #1: I think that is a quite well-conducted study.

I think that it is important in the conclusion part to explicit in which dimensions of the outcome edoxaban perform better than other DOAC, also because this is a founded study and I think that it is more ethical. 

The novelty of the paper is very low, but real-word data in DOACs are always appreciated.

Answer: There is a multitude of articles prior to ours presenting the incidence of serious complications of DOACs in real life. In this sense, it is true that the novelty is scarce. However, there are hardly any publications specifying that the clinical results come from rigorous clinical protocols. In our opinion, the novelty of our work consists on showing that, despite carrying out an intensive and specialized protocol, the incidence of complications is considerable. This allows us to take a critical view of many of the existing results, which could present underestimations in the rate of major complications. Therefore, it can cause excessive misguided optimism regarding the results of DOACs in routine clinical practice.

Reviewer #2: Authors present a prospective, observational, single center study enrolling over 600 patients with atrial fibrillation (MACACOD study) addressing the issue of efficacy and safety of DOAC in a real-world setting. Below my major and minor concerns:

MAJOR

• The paper begins as an observational prospective single center study (title, abstract, materials and methods, results) and ends as a non-systematic review of the literature (Discussion). I think it goes beyond the scope of an observational study to comment on the whole literature present. Furthermore, no mention has been made on how some studies have been selected and some others have been discarded

Answer: We prepared an extensive review of the literature to demonstrate the wide variability of clinically relevant outcomes (major complications, mortality) associated with DOACS in real life. As the reviewer points out, our review is not systematic. Although, the criteria for selecting articles were predefined: We selected publications on real-life data; registries from different geographical areas (Europe, America and, Asia) and with a significant number of patients. We also opted for studies with a methodology that suggests very reliable results (in the subjective but well-founded opinion of the authors), despite having a relatively modest number of patients. Finally, we also included some frequently cited articles. All this prolific published information has been used to compare with our results. This allows us to pose our main doubt regarding many of the previously published data, which, with high suspicion, are due to incomplete records or insufficient follow-up quality. This concern is always greater, the larger the sample size is!

On the other hand, we have registries from areas, e.g., Denmark, whose management results and epidemiological credibility have been recognized for the integrity and quality of their databases at the national level.

In the new version of the manuscript, we have made the selection of the compared works more explicit. See in Discussion: 

 “…The incidence of mortality was 6% p.y. We have used these figures to compare our results with other studies in Tables 4 and 5. 

The selection of these comparative examples does not come from a systematic review. The criteria to select the publications are as follows: first, pivotal studies of the 4 DOACs currently used in the real world; second, real-life results from different geographical areas (Europe, America, and Asia); then, registries with a very high number of patients or registries with a methodology that suggests highly reliable results regardless the number of patients enrolled (in the subjective opinion of the authors). In addition, we included frequently cited publications. We believe that our selection is representative of the literature on DOACs in the real world.”

• Discussion section should focus instead on interpretation of the interesting results that you presented in the paper, trying to pinpoint what is relevant to the reader and why, and underlining the eventual limitation and possible bias. You might of course compare your results to those of other studies, but Table 4, 5 and 6 go beyond this scope.

Answer: We understand the reviewer's comment, and it was one of the points that caused the most discussion in our group. However, we are now convinced that Tables 4-6 are very useful to fulfill one of the objectives of the work, already reflected in the title: Critical appraisal of real-world data.

To our knowledge, there is no publication that, in addition to present quality clinical results (as we think ours are), proposes a reflection of this type on the pre-existing results in real life. For this reason, we believe that it is necessary for the publication to maintain the original proposal for Tables and Discussion.

• Edoxaban and Apixaban population in this study are very different (percentage of naïve patients, antiplatelets drugs, previous bleeding, previous stroke). When presenting non pre-specified, subgroup analysis comparing two very different populations care must be taken not to over-emphasize differences that could easily be explained by the presence of a bias. Such results could be at most “hypothesis generating” and certainly can not support that “In our experience, edoxaban appears to perform better than other DOACs”. In order to state this, you need a RCT or at least a patient level meta-analysis of the available RCTs.

Answer: Thank you for the comment. We agree with your point of view, so the sentence in the manuscript is now:

“In our results, it seems that the edoxaban group performs better than other DOACs, as shown in some propensity-score real-life nationwide data from Lee [6], Maston [57], and Paschke [58]. Otherwise, our DOAC groups are not equal in terms of ages or basal risks, so these differences in co-variates could explain the observed differences in outcomes. More RCT or at least patient-level meta-analysis are necessary to evaluate the differences in safety and effectiveness between DOACs.”

MINOR

• In the section “Materials and methods - Blood tests” authors described meticulously how FXa activity was monitored; by the way, no mention is present in the results section about such an experiment

Answer: Thank you for the comment. We have added the trough and peak anti-FXa activity levels and its difference in Table 1. Also, we have included anti-FXa levels in the evaluation of the risks of major thrombotic, major hemorrhagic and all bleeding complications in edoxaban patients in Table 3. There are some sentences added to the manuscript:

In the results:

“We also found significant differences between edoxaban and apixaban in anti-FXa activity trough level (0.10 and 0.62 UI/Ml, P<0.01) and the trough-peak difference (1.14 and 0.67 UI/ml, P<0.01).

Between patients taking both edoxaban doses, those on the 30mg were older (81.4 vs 75.9 years), with higher CHA2DS2-VASc score (4.42 vs 3.69) and with worse mean CrCl (44.3 vs 74.3 mg/dL) but without statistically significant differences except for anti-FXa activity trough, peak and its difference (view Table 1).”

“Lower hemoglobin was found to be a significant risk factor for both hemorrhagic (HR 0.97, 95%CI:0.95-0.99, p=0.017) and all bleeding complications (HR 0.99, 95%CI:0.97-1.00, p=0.047) and elevated anti-FXa activity trough (pre-dose) level for all bleeding complications (HR 3.73, 95%CI:1.38-10.08, p=0.009).”

In the discussion: 

“Risk factors related to bleeding were low hemoglobin, anti-FXa activity trough (pre-dose) level and history of previous bleeding, but not age.”

• Results are often described as statistically significant without reporting confidence intervals and p-value. Even though these are present in the tables, they should be present also in the results section.

Answer: Thank you for the comment. We have added the confidence intervals and p-value in the results section.

---

## [Decision Letter · Decision Letter 1]

5 Dec 2022

Clinical outcomes in patients with atrial fibrillation treated with DOACs in a specialized anticoagulation center: Critical appraisal of real-world data

PONE-D-22-21794R1

Dear Dr. Moret Puig,

We’re pleased to inform you that your manuscript has been judged scientifically suitable for publication and will be formally accepted for publication once it meets all outstanding technical requirements.

Kind regards,

Giulio Francesco Romiti

Academic Editor

PLOS ONE

Reviewers' comments:

Reviewer #2: Review process improved the paper. Authors answered all the comments and extensively reviewed the paper, that is now, in my opinin, suitable fo publications.

---

## [Editor Report · Acceptance letter]

14 Dec 2022

PONE-D-22-21794R1 

Clinical outcomes in patients with atrial fibrillation treated with DOACs in a specialized anticoagulation center: Critical appraisal of real-world data 

Dear Dr. Moret Puig:

I'm pleased to inform you that your manuscript has been deemed suitable for publication in PLOS ONE. Congratulations! Your manuscript is now with our production department. 

Kind regards, 

on behalf of

Dr. Giulio Francesco Romiti 

Academic Editor

PLOS ONE